# Coffee, Farmers, and Trees—Shifting Rights Accelerates Changing Landscapes

**Claude A. Garcia** [1,2,*] , **Jérémy Vendé** [3] , **Nanaya Konerira** [2,4,5] , **Jenu Kalla** [2,5] , **Michelle Nay** [2] ,
**Anne Dray** [2,6] , **Maëlle Delay** [2] , **Patrick O. Waeber** [2] , **Natasha Stoudmann** [2] , **Arshiya Bose** [2] ,
**Christophe Le Page** [7] , **Yenugula Raghuram** [8] , **Robert Bagchi** [9] , **Jaboury Ghazoul** [6] ,
**Cheppudira G. Kushalappa** [4] **and Philippe Vaast** [10,11]

1. CIRAD, UPR Forêts et Sociétés, F-34398 Montpellier, France
2. ETH Zurich, Institute of Terrestrial Ecosystems, Forest Management and Development (ForDev),
   8092 Zurich, Switzerland; nanayakm@yahoo.com (N.K.); jenu.gowda@gmail.com (J.K.);
   michelle.nay@usys.ethz.ch (M.N.); anne.dray@usys.ethz.ch (A.D.); delay.maelle@gmail.com (M.D.);
   patrick.waeber@usys.ethz.ch (P.O.W.); n.stoudmann@hotmail.com (N.S.); arshiyabose@gmail.com (A.B.)
3. AgroParisTech, F-34093 Montpellier, France; jeremy.vende@agroparistech.fr
4. Ponnampet College of Forestry, University of Horticultural and Agricultural Sciences, Bizarre Nagara,
   Ponnampet, Karnataka 571216, India; kushalcg@gmail.com
5. French Institute of Pondicherry, 605001 Pondicherry, India
6. ETH Zurich, Institute of Terrestrial Ecosystems, Ecosystem Management, Universitätstrasse 16,
   8092 Zürich, Switzerland; jaboury.ghazoul@env.ethz.ch
7. CIRAD, UPR Acridologie, F-34398 Montpellier, France; christophe.le_page@cirad.fr
8. Central Coffee Board, Devaraj Urs Rd, opposite Vishveshvaraiah Tower, Ambedkar Veedhi, Vasanth Nagar,
   Bengaluru, Karnataka 560001, India; raghuram.yenugula@gmail.com
9. Department of Ecology and Evolutionary Biology, University of Connecticut, Storrs, CT 06269, USA;
   robert.bagchi@uconn.edu
10. CIRAD, UMR Eco&Sol, University of Montpellier, F-34000 Montpellier, France; philippe.vaast@cirad.fr
11. ICRAF, Hanoi 100000, Vietnam
\* Correspondence: claude.garcia@cirad.fr; Tel.: +41-44-6328975

**Abstract:** Deforestation and biodiversity loss in agroecosystems are generally the result of rational choices, not of a lack of awareness or knowledge. Despite both scientific evidence and traditional knowledge that supports the value of diverse production systems for ecosystem services and resilience, a trend of agroecosystem intensification is apparent across tropical regions. These transitions happen in spite of policies that prohibit such transformations. We present a participatory modelling study run to (1) understand the drivers of landscape transition and (2) explore the livelihood and environmental impacts of tenure changes in the coffee agroforestry systems of Kodagu (India). The components of the system, key actors and resources, and their interactions were defined with stakeholders, following the companion modelling (ComMod) approach. The underlying ecological processes driving the system were validated through expert knowledge and scientific literature. The conceptual model was transformed into a role-playing game and validated by eight workshops with a total of 57 participants. Two scenarios were explored, a No Policy Change as baseline, and a Restitution of Rights where rights to cut the native trees are handed over to farmers. Our results suggest that the landscape transition is likely to continue unabated unless there is a change to the current policy framework. However, the Restitution of Rights risks speeding up the process rather than reversing it, as inter alia, the differential growth rate between exotic and native tree species, kick in.

**Keywords:** companion modelling; agroforestry; *Grevillea robusta*; India; policy; role playing games

## 1. Introduction

### 1.1. Problem Framing

Human agency now rivals natural processes in shaping and driving the Earth system to an extent that many advocate the recognition of a new geological era, the Anthropocene [1]. This is particularly true of agroecosystems where, from the local to the global, the human enterprise affects directly or indirectly most if not all elements and processes [2,3]. One of the outcomes of this stranglehold is that several proposed planetary boundaries that define the safe operating space for humanity have already been crossed, in particular the core Biosphere Integrity boundary (formerly referred to as Rate of Biodiversity loss) [4–7]. Conservation efforts aim at halting or reversing this trend. Considering that just under 15% of the globe's 13.4 billion ha of land surface is devoted to conservation, against 11% to arable land and permanent crop production [8,9] and that further extension of protected areas will be costly even if at all feasible, biodiversity conservation should increasingly be contemplated outside protected areas [10–12].

Since human agency is of primary importance in shaping these landscapes [13], mainstreaming biodiversity conservation into agroecosystems requires careful consideration of the values, needs, constraints, and actions of farmers and other relevant stakeholders as primary drivers of change [14,15]. Values and actions can be contradictory, and their elicitation is not straightforward. In Indonesia, for example, farmers readily change their livelihood system by replacing forests and agroforestry systems with alternative crops such as oil palm in response to market incentives, despite their cultural attachment to forest [16]. In many cases, deforestation and the loss of biodiversity are the result of rational choices, not of a lack of awareness or knowledge [17]. Failure to adequately represent agency can lead to policies that are little more than wishful thinking, or even potentially exacerbating negative outcomes [18,19].

Agents (sensu [18]) seeking to improve agroecosystem governance are confronted with seemingly intractable difficulties. (i) They need to take into account the bounded rationality of the stakeholders, the fact that their rationality is limited by the information they possess, their cognitive limitations, and the finite amount of time they have to make a decision [20]. They need to take into account the fact that drivers of decision might be tacit—so obviously nobody mentions them to outsiders—cryptic, unknown to all but the most discerning observer, or even concealed—such as freeriding or corruption [21,22]. (ii) The system they deal with is highly unpredictable, with complex and nonlinear interactions, feedback loops and delayed effects between individual decisions, collective behaviour and ecological processes [23]. (iii) Finally, the system is highly adaptive and stakeholders can quickly exhibit behavioural plasticity, finding loopholes or revising their strategies.

Models can help explore this complexity [23–25], provided they capture the full complexity of the stakeholders' decision-making process. Dismissing this critical system feature and behaviour limits the model's validity and outputs [26]. Thorough and recent reviews of participatory approaches with an emphasis on collaborative modelling exist, and can be referred to [27,28]. Participatory models [29], agent-based modelling [23], and scenario planning [30] have been proposed as possible avenues to help agents 'muddle through' this quagmire.

### 1.2. Case Description

The district of Kodagu in the Western Ghats (India) is located in a major biodiversity hotspot [31]. Farmers in Kodagu produce one third of Indian coffee [32,33], in addition to rice and other cash crops such as cardamom (*Eletaria cardamomum* M.), ginger (*Zingiber officinale* R.), and areca nut (*Areca catechu* L.) [34]. The resulting landscape is a dynamic mosaic of terraced rice fields, evergreen and moist deciduous forest fragments, some of them sacred [35,36], and coffee farms. These coffee farms, locally referred to as Estates, produce mostly Robusta coffee (*Coffea canephora* var *robusta* L.), under complex, multi-storeyed agroforestry systems [17,37]. A survey of more than 21,000 trees in the landscape identified more than 270 different tree species [38]. The landscape is also dotted with water

storage tanks used to irrigate coffee to induce flowering, increase yield and reduce the need for dense shade cover [17,39].

Land use and land cover change, and the consequent loss of biodiversity in this landscape, result primarily from the decisions individual coffee farmers make on their estates. Time series have shown that most changes to the tree cover happen inside privately owned land, leaving the area under the direct control of the government and the Karnataka Forest Department largely untouched [17]. Tree cover changes mainly because of three different management interventions: when coffee farmers (i) replace forest habitats with coffee plantations, (ii) remove shade trees to increase coffee productivity, and (iii) replace a rich and diverse evergreen and moist deciduous canopy cover by planting Silver Oak (*Grevillea robusta* A. Cunn.), a fast growing tree originating from Australia [40,41]. Of these three practices, the slow conversion to Silver Oak has received most attention, and the reasons advanced by the coffee farmers are well documented [40–42]. This species can be logged and traded easily, as it lacks any specific protection rights. This is not the case for native species, whose harvest is restricted and for which full value of the timber cannot be recovered by the coffee farmer [17].

For over 10 years, coffee farmers and their representatives have been actively advocating a policy change, demanding full ownership rights over all the trees on their land, not only the exotic ones, and the lifting of restrictions on the harvesting of native trees [42,43]. Over the years, this issue has led to considerable friction between coffee farmers and the Karnataka Forest Department which opposes farmer demands for tree ownership rights out of concern of the environmental impact this would have on tree cover across the district. Vocal representatives of the coffee farmers, for example the Codagu Planters Association (sic), deny that the granting of tree rights would result in a loss of tree cover or conversion [44]. This position resonates with discourses on devolution, where privatization and downsizing of state control on one hand, and community empowerment on the other, have replaced earlier State-control centred narratives [45]. This polarized debate has led to a long-lasting standoff between the opposing parties, farmers demanding rights to cut trees, and the Karnataka Forest Department opposing this, representing an all-too classical conflict situation between conservation and development [17,46,47].

### 1.3. Objective and Approach

The research we present here in Kodagu was initiated in 2009, as a sub-component of the Connecting, Enhancing and Sustaining Environmental Services and Market Values of Coffee Agroforestry in Central America, East Africa And India (CAFNET) project, that lasted from 2007 to 2010. Other components of the project described landscape and its components, including the distribution of biodiversity, market trends and stakeholder strategies that have been presented in the previous section. To (i) better understand the drivers of decision making by coffee farmers and explore their coping strategies when facing a policy change, and (ii) contribute to the resolution of this standoff, we developed a three-step participatory process using Companion Modelling [31], building upon our previous research in the area [48]. This adaptive process started in 2009 and continued until 2012, bringing together academics, representatives of the Central Coffee Board of India and of the Karnataka Forest Department, local NGOs, private coffee trading companies, and representatives of coffee planters' associations.

We selected the approach as it is useful to promote dialogue, helps stakeholders understand different viewpoints and is flexible enough to explore a wide range of human behaviour—from very conservative to very disruptive ones [49]. We combined participatory modelling, the use of boundary objects and role-playing games, and socio-economic scenario definition and exploration, following the Companion Modelling (ComMod) approach [50–52]. We discuss how this approach allowed us to explore hidden or often overlooked drivers of landscape change, and how it unveiled surprising behaviour. We also identify the limits of the approach. This will advance the theoretical grounding behind the use of games in scenario planning and natural resources management.

## 2. Materials and Methods

### 2.1. Companion Modelling

The model and the game were developed through the approach called Companion Modelling (ComMod) [53]. ComMod is an interactive process where stakeholders, supported by the ComMod practitioners, develop models and simulations. Both are then used as mediating tools in the support of problem framing, collective learning, dialogue, and collective decision-making [54]. A ComMod process undergoes multiple iterations, where the problems, models, and underlying understanding of the process at play are progressively refined and if necessary redefined—end-user interests drive the modelling and simulation activities [50]. We followed in the entire process the charter of the Companion Modelling [55].

The initial phase took place between March and August 2009, with 68 semi-structured interviews, participant observation (5 months of presence in the field by one of the authors), 3 group discussions and modelling workshops and 5 game sessions organised, overall totalling 72 stakeholders, essentially coffee farmers but also academics, timber merchants and representatives of the Karnataka Forest Department [56]. This led to the design of the conceptual model and the validation of the first game prototype (Figure 1).

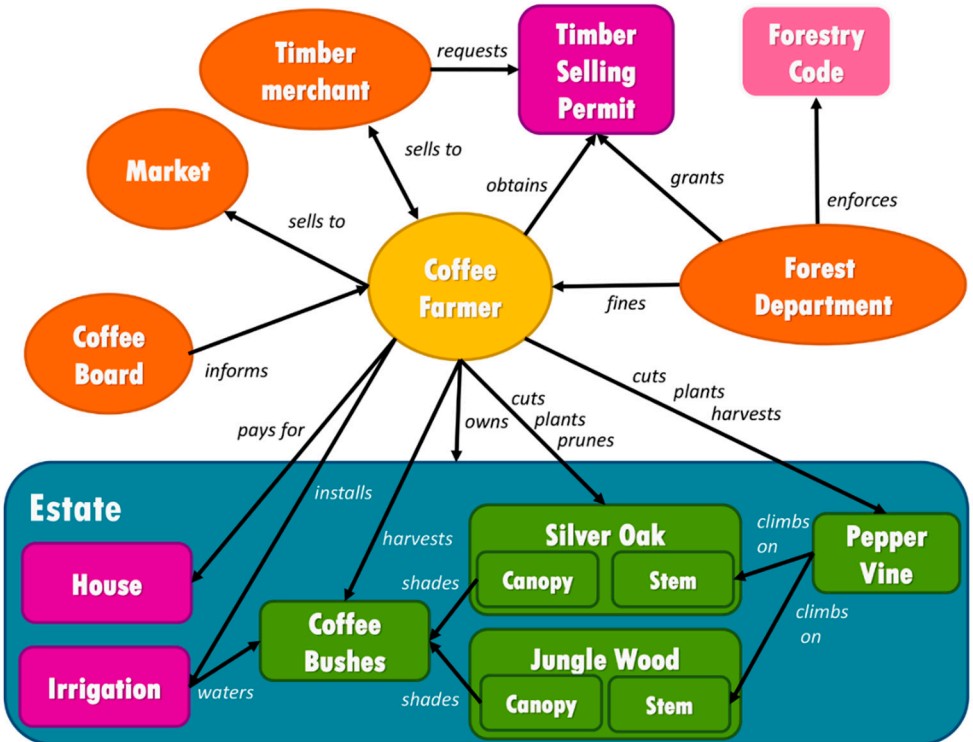

**Figure 1.** Conceptual model of the Kodagu role playing game—The system components are either Resources (round squares) or Actors (ovals). Resources are spatial units (blue), physical elements (violet), sets of norms (pink) or natural elements (green). The Actors are either autonomous (yellow) or scripted (orange), meaning that, in our model, their behaviour is restricted. In practice, orange actors were played by members of the research team. Actors and Resources are connected by one or two-sided arrows that represent the main interactions between them. Each arrow can be understood as a verb linking two components to create a sentence. For example, a *Coffee Farmer* plants a *Silver Oak.* For the sake of clarity, we have omitted the interactions between *Coffee Farmer* and *Jungle Wood*, identical to the ones between *Coffee Farmer* and *Silver Oak.*

The two scenarios were defined in a workshop of the CAFNET project involving academics, representatives of the Central Coffee Board, officials from the Karnataka Forest Department and coffee farmers. Previous research [17,41] had identified tree tenure rights as critical to the decisions of coffee

farmers. Both scenarios use the same model but with a slightly different set of rules regarding the possibilities of *Coffee Farmers* to cut, transport and sell *Jungle Wood* to the *Timber Merchant*. The first scenario (No Policy Change—S1) represents the situation as it currently is in the field: coffee farmers are free to cut and sell the silver oaks in their estate, but have to request a specific permit from the Forest Department to sell their jungle wood. The research assistant playing the *Forest Department Officer* during gaming sessions behaves in such a way that securing this permit is difficult and time consuming, mimicking reality. The second scenario (Restitution of Rights—S2) represents the situations as advocated by representatives of the coffee farmers' associations: no restrictions on the selling of *Jungle Wood* or *Silver Oak* from their *Estate*. In both cases, at the request of the stakeholders involved in the game design, we introduced a coffee market crash on Turn 4, mimicking recent events at the time in the global coffee market. Other scenarios like payment for ecosystem services, landscape labelling or the strengthening of the restrictions on timber felling were all contemplated and dismissed on the grounds of their perceived implausibility.

The game was then brought back to the field for eight exploratory game sessions involving 57 players organised between November 2010 and May 2011. Players for these sessions were recruited from the list of farmers involved in the CAFNET project. They were contacted individually and invited to participate in a workshop to play a game that explores how trees are managed in coffee estates in Kodagu. The nature of the game requires active consent by the participants. In the opposite case, they would simply not turn up on the day. This happened regularly and explains why we have a variable amount of participants for each session. Similarly, participants were free to leave at any moment. Departures mid-game happened in only three occasions, forcing us to exclude the observations from the data analysis. No stipend or compensation was offered beyond the lunch provided, to avoid generating extrinsic motivation. Recruiting players for the initial sessions was difficult, the concept of playing a game being unusual to explain. Once word of mouth had spread, it became easier, and the last session was organised at the request of the participants themselves. Keeping the number of participants low ensured we could adequately manage the flow of the game.

### 2.1.1. Model, Game Rules and Sessions

We will use capitals and italics when we refer to model components (*Estate, Coffee Farmer*), in order to distinguish them from elements of the system (estate, coffee farmer).

The conceptual model is the first output of the ComMod process. It was constructed to answer the question "What drives tree management in the coffee estates of the district of Kodagu?" (Figure 1). Central to the model is the link between coffee yield, shade level and the presence of irrigation. The shape of this interaction was discussed with the coffee farmers and its equation defined based on scientific literature [57], allowing us to merge scientific knowledge with the coffee farmers' empirical knowledge. The management practices identified by coffee farmers as being important were represented as interactions between the *Coffee Farmer* and the resources on his *Estate*, and the other actors of the system. The model was co-constructed with farmers over a period of nine months. It evolved considerably during this period, to accommodate for the feedback received by participants at each step of the process, as described in the following section.

The conceptual model, though simplified, has 5 different actors, handling 13 different resources, and more than 19 interactions. To allow players to handle this complexity, we produced a toy model translating the conceptual model into a boardgame that could be played with coffee farmers, academics and other stakeholders (Figure 2). Both the conceptual model and the boardgame act as boundary objects [58].

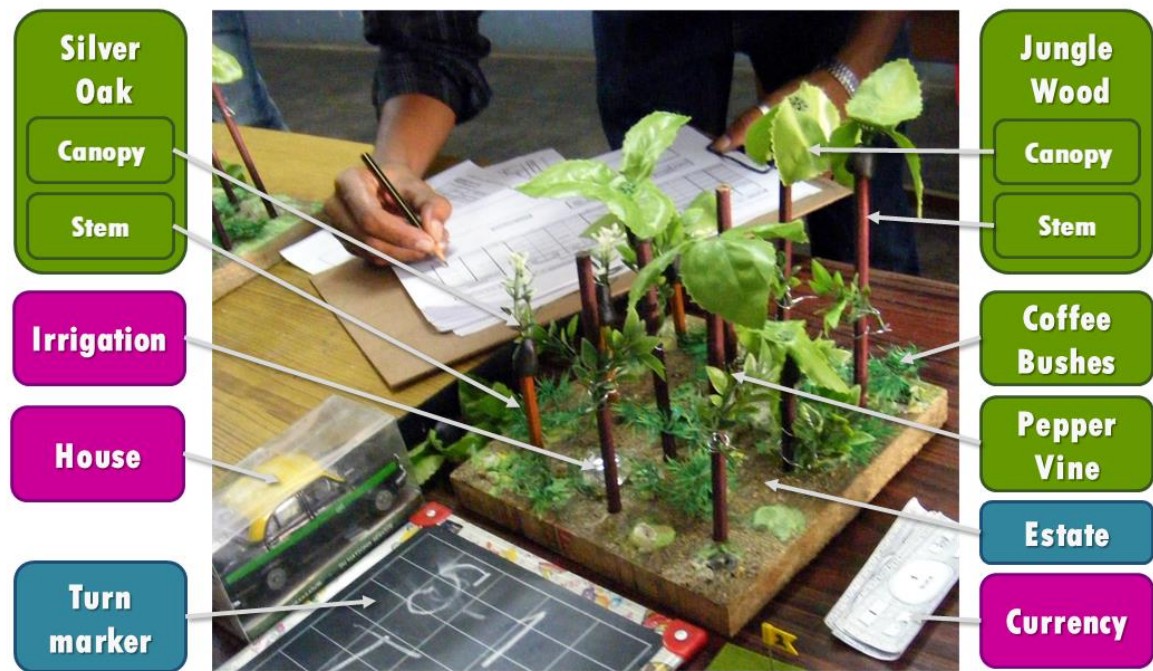

**Figure 2.** Creating a toy model. In order to allow stakeholders to understand, manipulate and explore the behaviour of the conceptual model, we transformed it into a physical toy model. All the Resources identified in the conceptual model were represented. The Actors became players of the game, and the Interactions were detailed as the rules of the game. In addition to the resources listed in Figure 1, we used fake bank notes to act as currency and allow for economic transactions during the game sessions.

An implicit reality was applied, i.e., a simplification of actors, resources, and space were used in the concept and corresponding game. Thus, the toy model represents an instanciation of the conceptual model. Taken together, the tokens, the board, the rules, the research team, the players and the physical location where the game session is run constitute an embodiment of the conceptual model presented earlier (Figure 1). Through the process of playing, the participants explore the behaviour of the model, gather information, make choices, observe the outcomes of their actions and adjust their strategies accordingly. The research team can gather data during this process via three channels: (1) Documenting the decisions made and their outcomes—for example, counting the trees left in the estates, or recording coffee yields; (2) observing the social interactions and documenting the narratives developed by the players; and (3) gathering direct feedback from the participants, during the debriefing.

Each player is given a board representing the *Estate* with 10 holes filled initially (at round 0 of the game) with toy models of six *Jungle Wood (JW) Trees*, two *Silver Oak* (*SO*) *Trees* and two *Jungle Wood* seedlings. The collection of *Estates* constitutes the gamescape. Players need to manage their *Estate* by:

- cutting and pruning Trees
- obtaining and planting SO Seedlings and Pepper Vines
- buying and installing Irrigation
- selling Coffee, Pepper and Timber
- interacting with players and officers
- paying their living costs

There are several offices set up in the game room, where players can obtain information and talk to the officers. In the *Clerk's Office*, the market prices of *Coffee*, *Pepper* and *Irrigation* are displayed. The prices of *Timber* are displayed in the *Timber Merchant Office*, and forms for obtaining *Timber Transport Cutting and Sales* (TTCS) certificates for selling *Jungle Wood* are available from the *Forest Department*

*Office*. The *Coffee Board Office* shows graphs of the dependency of *Coffee* and *Pepper* productivity on shade and *Irrigation*. *Silver Oak Seedlings* and *Pepper Vines* (one per tree) can be bought at the *Nursery*, while *JW Seedlings* appear naturally in the *Estate*.

All the elements of the conceptual model were used to create the game. The resources became the tokens (*Silver Oak Stem, Silver Oak Canopy, Irrigation, Pepper Vine, Selling Permits,* etc.) or the board itself (the *Estates*). The actors whose strategies we were interested in became the roles played during the game—*Coffee Farmers* (Figure 3). Other relevant actors (*Officers*) were represented by members of the research team, and their behaviour was scripted according to the common understanding of the actor by the participants of the design process. Finally, the processes and interactions became the rules of the game, dictating what should happen when, in which order, and what options the players had at their disposal. The ecological processes form the core engine of the game—natural regeneration and yield—dictating the outcome of the interactions between the joint decisions of the *Coffee Farmers* and the physical constraints we had described in the model. Risk in the model comes in three forms. The first one is inbuilt in the random factor included in the productivity function. The other is the market crash scripted on turn 4. The third one is the possibility of being caught by the *Forest Officer* while transporting *Timber* without permit.

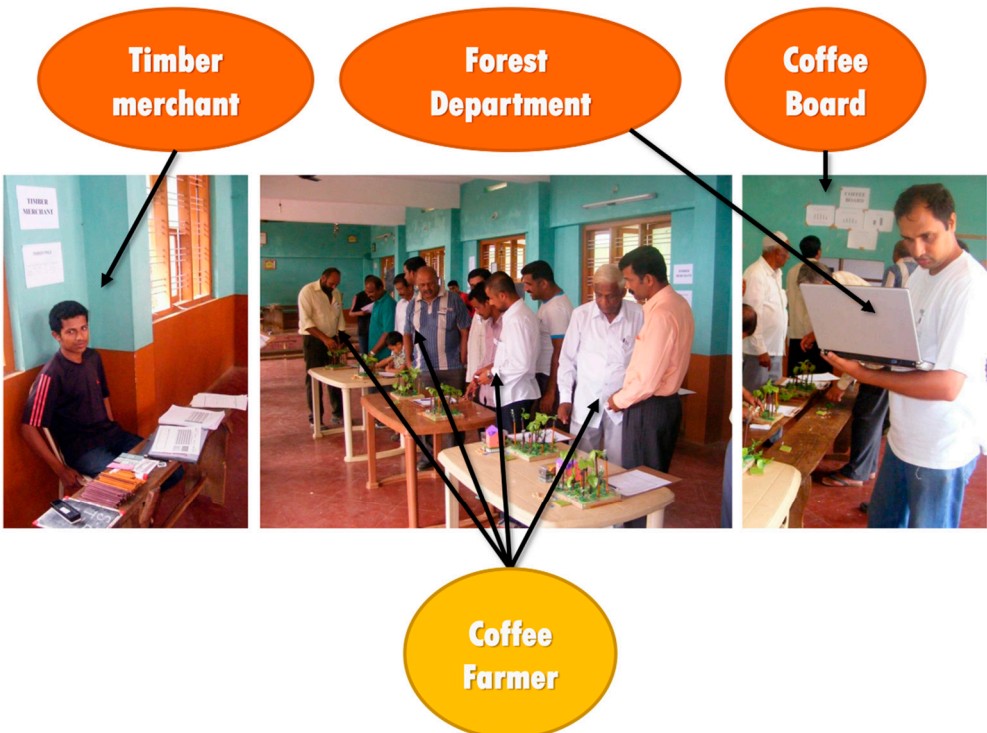

**Figure 3.** Playing the game. The combination of the board, the tokens, the players, the research team impersonating the scripted actors, the set of rules that define the steps and the outcomes of the actions taken, and the room where the game is played constitute a incarnation of the conceptual model, allowing the participants to manipulate and explore the behaviour of the model and the researchers to record the outcomes and generate discussions.

Unlike commercial games where victory conditions are given to the players, nothing here was imposed except the need to achieve a mininum income of 80 CINR (game currency, for CAFNET Rupee) to cover recurring costs. This threshold was low enough that it would easily be fullfilled with the sole income from coffee. Doing so ensured none of the participants would waste their day due to early elimination from the game. Beyond that, players were free to decide what they wanted to aim for. The ensuing discussions allow the research team to elicit these "victory conditions", and, through them, discuss the values and objectives that drive the changes in the landscape.

The two scenarios tested involve different sets of rules regarding the sale of timber. In S1 (No Policy Change), there are two channels to sell *Jungle Wood*. Players can sell their *Jungle Wood* logs to the *Timber Merchant* and receive the full value of the timber through the proper procurement of a valid *Timber Selling Permit*. This requires filling in forms, obtaining stamps and signatures from several members of the research team posing as administrative clerks, including the *Forest Department Officer*, who all have to deal with the simultaneous demands of all prospective timber sellers, under a strict time constraint. This process is difficult, time consuming and frustrating, according to the players themselves. At the request of the stakeholders involved in the game design, as the game progresses—more rounds go by—players could also try to secure a direct agreement with the *Timber Merchant* who would do the administrative paperwork on their behalf, at the cost of a fraction of the timber price. In S2 (Restitution of Rights), there is no longer need for the *Timber Selling Permits* and the players can bring their *Jungle Wood logs* directly to the *Timber Merchant* and receive the full payment as per the market price. Both scenarios are played in the same day, S1 in the morning and S2 in the afternoon. In both cases, at the start of the fourth round, a coffee market crash is announced by the game master, which means that the price for coffee bags will drop from 6 to 2 CINR for this turn only.

*Trees* can be cut, by removing the toy model from its hole in the *Estate*. A *Tree* that is cut during the game by a player cannot be replaced back. Instead, a new seedling will need to appear (*JW*) or be planted (*SO*). A *Tree* can also be pruned by removing the *Crown* and leaving the *Stem* in place. From round to round, the *Trees* grow and the *Crown* reestablishes. All *Seedlings* are replaced with *Saplings* at various rates depending on the species: *SO Saplings* become mature *Trees* after one round, while *JW Saplings* need two rounds to become mature. *Crowns* regrow every round. At the end of every round, *Farmers* collect their crop yields, which vary depending on their management decisions and a random factor.

The shade–yield relation [57] was taken as basis for the model's dynamics. The parameters were fitted to simulate outcomes judged plausible by the participants and adjusted through multiple testing rounds. Shade is calculated for every Estate with every Sapling and mature Tree with Crown adding 10% of shade to the Estate. Equations (1)–(3) were used to calculate the coffee and pepper yields. The Coffee yield YC (in bags) depends on Shade (S) values of the Estate (from 0 to 1), and $\alpha$, $\beta$, $\delta$ a set of parameters that change with the presence of Irrigation (Equation (1), Table 1):

$$Y_C = \alpha S^2 + \beta S + \delta + \varepsilon(S) \tag{1}$$

$\varepsilon(S)$ (Equation (2)) is a random factor that depends on Shade, to represent the dampening effect shade has on yield fluctuations [57]. With K following a normal distribution, values were rounded to the integer and given a minimum value of 5 bags to allow unlucky players to remain in the game:

$$\varepsilon(S) = 10K(1-S) \tag{2}$$

*Pepper yield* $Y_P$ (in bags) depends on: $V_{Sh}$ (Shade), the number of *Pepper vines* growing on *Trees* with *Crowns* and $V_{FS}$ (Full Sun) the number of *Pepper vines* growing on pruned *Stems* (Equation (3)). The value is rounded to the closest integer:

$$Y_P = V_{Sh} + 0.6V_{FS} \tag{3}$$

**Table 1.** Parameters of the yield production function with and without irrigation.

| Parameter | Without Irrigation | With Irrigation |
|:---:|:---:|:---:|
| $\alpha$ | −45 | −35 |
| $\beta$ | 40 | 20 |
| $\delta$ | 10 | 20 |

If players do not sell their harvest, it is lost since there is no storage capacity. *Coffee Farmers* can interact with each other to sell and buy goods. The *Forest Department* officer can control players "transporting" (walking around with) or cutting *Trees* to ensure that they have a *Certificate*, and can eventually fine them. Additional costs and values using throughout the game are listed in Table 2.

**Table 2.** Additional costs and values.

| Item | Cost (in CINR) |
|---|---|
| *Irrigation* (per round) | 10 |
| *Pepper vine* | 1 |
| *Silver Oak Seedling* | 1 |
| Simple House (per round) | 80 |
| Medium House (per round) | 140 |
| Large House (per round) | 220 |
| *Jungle Wood Sapling* | 0 |
| *Jungle Wood* mature *Tree* | 16 |
| *Silver Oak Sapling* | 3 |
| *Silver Oak* mature *Tree* | 10 |
| Coffee bag (normal price) | 6 |
| Coffee bag (crash price) | 2 |
| Pepper bag | 2 |

The following values are recorded every round for each player's estate in an Excel sheet: (i) Percent shade, (ii) Irrigation, (iii) number of Pepper Vines (sun/shade), (iv) market gains (from coffee & pepper bags, SO Saplings, SO mature Trees, JW mature TTSC, JW mature direct), (v) size of house (simple/medium/large), (vi) expenses (*Pepper vines* & *SO Seedlings* bought, fines, misc. costs), and (vii) tree abundance (*SO/JW*, *Seedling*, pruned *Sapling*, crowned *Sapling*, pruned mature *Tree*, crowned mature *Tree*). These data are then used to calculate yields and then players' income depending on market prices.

### 2.1.2. Scenario Testing

The RPG (role playing game) was played in eight playing sessions, each player playing the two scenarios S1 (No Policy Change) and S2 (Restitution of Rights). Fifty-seven players (46 males, 9 females) participated in total and numbers varying between sessions (min 5, max 8). The age of the participants ranged between 19 and 73 years, with a median of 46.5. The literacy level was high, 33 of the participants holding degrees above a Bachelor's. Both scenarios began with the same initial conditions, described in the Methods section. We measured and recorded the following indicators during the game sessions, which also served to support the discussion with the players during the debriefing: (1) The income of the players, (2) the number of *Trees* sold in the market, (3) the number, (4) age class, and (5) species of the *Trees* left in their *Estate*. The cumulative outputs of the game sessions suggest the potential impacts of the policy change in terms of livelihood, monitored through the income of the players, the economy, represented by the supply of logs to the *Timber Merchant,* the environment, represented by the changes of the composition of the canopy cover of the *Estates*.

### 2.2. Statistical Analysis

We used mixed-effects models (using the lme4 v1.1-15 package in R v3.4.4, [59]) to examine the relationships between scenarios and turns and the various responses of interest (including revenues and tree cover). All models included scenario and round and their interaction as fixed-effects and variation among sessions and players was accounted for by adding random intercepts for players and sessions drawn from normal distributions. The models can be described as

$$\mu_i = \beta_0 + \beta_1 turn_i + \beta_2 scenario_i + \beta_3 turn_i \times scenario_i + b_{player_i} + c_{session_i}$$
$$b_j = Normal(0, \sigma_b); \; c_k = Normal(0, \sigma_c) \tag{4}$$
$$y_i \sim g(\mu_i, \; \sigma_\epsilon)$$

where $\mu_i$ is the predicted response for the $i^{\text{th}}$ observation. The $\beta_p$ are fixed effects parameters to be estimated and quantify the intercept ($\beta_0$), difference between turn 0 and each subsequent turn ($\beta_1$ is a vector with 3 elements), difference between scenarios 1 and 2 ($\beta_2$) and the interaction between turn and scenario (another 3 element vector, $\beta_3$). The terms $b_j$ and $c_k$ are the best unbiased linear predictors (BLUPs) describing the divergence of each player $j$ and session $k$ from the population mean, with both drawn from a normal distribution with mean 0 and standard deviation ($\sigma_b$ or $\sigma_c$) to be estimated. The third line describes the stochastic component of the model, where the recorded response for the $i^{\text{th}}$ observation is drawn from a distribution, $g(.)$, defined by $\mu_i$ and, when the distribution is normal, a standard deviation for the error, $\sigma_\varepsilon$. Models of total income, revenues from coffee, timber and pepper were fitted with linear mixed effects models assuming Gaussian error distributions. The proportions of tree locations occupied by any trees (total cover) and saplings and mature trees of jungle wood or silver oak were modelled using generalized mixed-effects models assuming binominal error distributions [60,61]. The full binomial models were examined for overdispersion. We assessed the statistical significance of each term by computing the log-likelihood ratio (LLR) of models with and without the term of interest. Tests of the main effects of scenario and turn were assessed after removing their interaction. Statistical significance (i.e., *p*-values) of differences between models was assessed by comparing the observed LLR to a null distribution, created using a parametric bootstrap performed in the pbkrtest package (v0.4-7, [62]). The null model (the model without the term of interest) was used to simulate the response data 999 times. Both the null model and the model with the term of interest were fitted to these simulated response data and the LLR between them calculated. The proportion of simulated LLR values that were greater than the observed LLR value was used as a test of the null hypothesis (that the term of interest was not important), providing the *p*-values presented in the results.

## 2.3. Ethics Statement

The research involved humans playing our games. No institutional board or ethics committee approved the study, but the researchers follow the Companion Modelling charter [55]. Consent was demonstrated by the participants attending the events and staying until completion. Data were processed anonymously and confidentiality ensured. The individuals in the pictures have given written informed consent to publish these pictures. Six research assistants of our research team were also members of that local cultural community, and the entire team was therefore aware of possible cultural differences and what is considered proper and respectful in the local context. The only issue that ever arose in the project was people asking why they had not been invited earlier. It is important to note here that there was no remuneration offered for the half-day time required to participate in the workshop; people from the local communities heard about the game and wanted to be part of it by free will knowing that they had the opportunity to talk freely without fear of any repression by anybody (e.g., authorities, forest service).

## 3. Results

### 3.1. Impact on Livelihoods

We recorded the total income of the players—adding the revenues from the sale of *Coffee, Pepper, and Timber* (Figure 4). The total income was significantly different between the scenarios with players having a 20 CINR (±4.3, LLR = 22.21, *p* < 0.001) higher income per round in S2. The difference came from the increased revenue from *Timber* (LLR = 125.61, *p* < 0.001), as there was no significant difference between the revenue from *Coffee* (LLR = 1.78, *p* = 0.179) or from *Pepper* (LLR = 3.44, *p* = 0.065). *Coffee Farmers* in the game were better off when the restrictions on the sale of *Jungle Wood* were lifted. The downward slope of the income curves reflects the fact that the game had a coffee market crash designed to happen in round 4, for both scenarios. It is of no relevance to the comparison to the two policy scenarios.

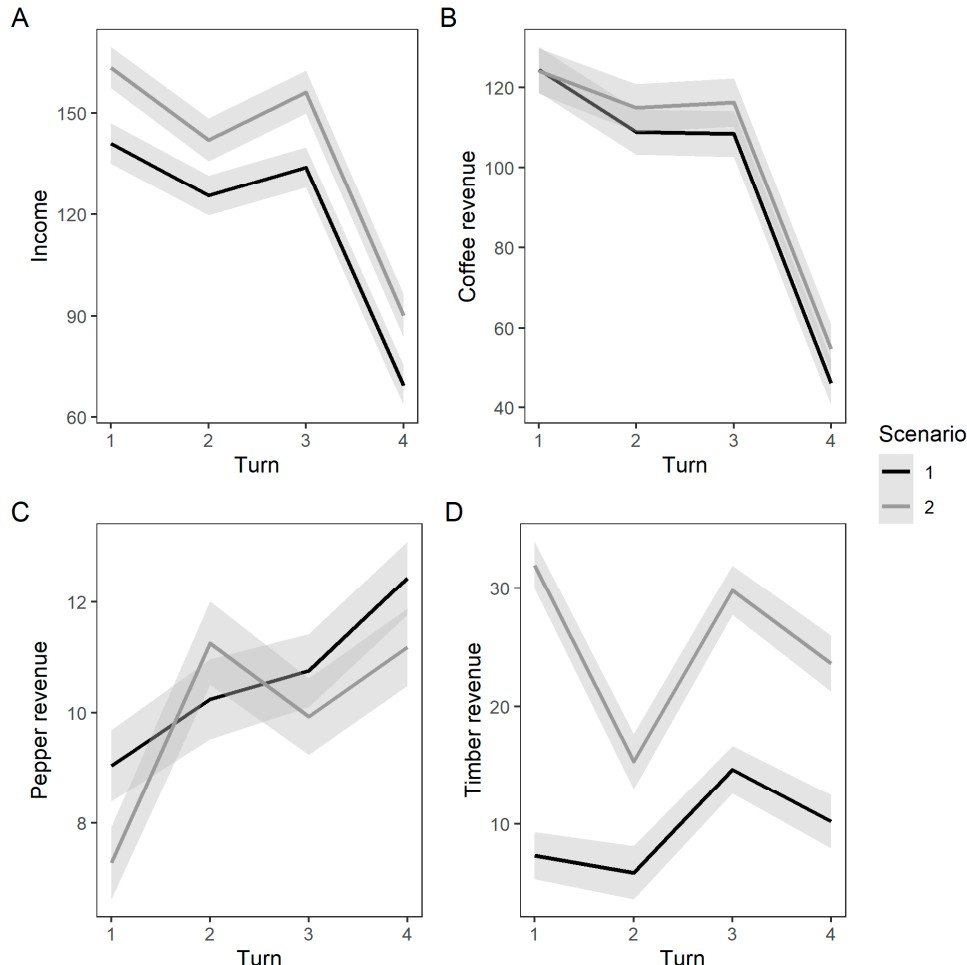

**Figure 4.** (**A**) income and revenue of the players from (**B**) coffee, (**C**) pepper and (**D**) timber. Income is in CINR, a fake monetary unit created for the purpose of the game. The market price for coffee in round 4 decreased because of the scripted market crash, reducing the income of all players. For all subfigures, the lines represent the predicted mean (Scenario 1 Black, Scenario 2 Gray) and the bands indicate the standard errors of the predicted mean.

### 3.2. Impact on the Timber Supply Chain

We monitored the amounts of *logs* sold to the *Timber Merchant* in both scenarios. In S1, players sold 234 logs in total, 76% of them *Silver Oak*. The two channels to sell *Jungle Wood*, with permit and through direct agreement, contributed 9 and 15% of the *logs* sold, respectively. In S2, the number of *logs* brought to the market more than doubled—517 *logs* in total, and 41% of them *Jungle Wood*.

### 3.3. Impact on Tree Cover

The evolution of the tree cover of the *Estates* in the game was the result of interactions between the rules in the game representing ecological processes and the management decisions of the *Coffee Farmers*. Tree cover could be tracked with three indicators: (1) Tree density (number of *Trees* on the boards), (2) stand structure (proportion of seedlings, saplings and mature *Trees*) and (3) stand composition (proportion of *Jungle Wood* and *Silver Oak*).

The average number of *Trees* (saplings and mature) per estate was significantly greater under scenario 1 (7.63 ± 0.13 trees) than under scenario 2 (6.22 ± 0.14 trees; LLR = 91.02, $p < 0.001$; values after ± are Standard Deviations). The average density of *Jungle Wood Trees* was also greater in scenario 1 (5.55 ± 0.38) than scenario 2 (3.66 ± 0.38; LLR = 143.45, $p = 0.001$). In contrast, the density of *Silver Oak Trees* was slightly, but significantly, greater in scenario 2 (2.56 ± 0.35) than scenario 1 (2.09 ± 0.34;

LLR = 14.14, *p* < 0.001). In both scenarios, the density of *Silver Oak* showed an upward trend as the rounds progressed, while the density of *Jungle Wood Trees* remained stable or decreased (Figure 5).

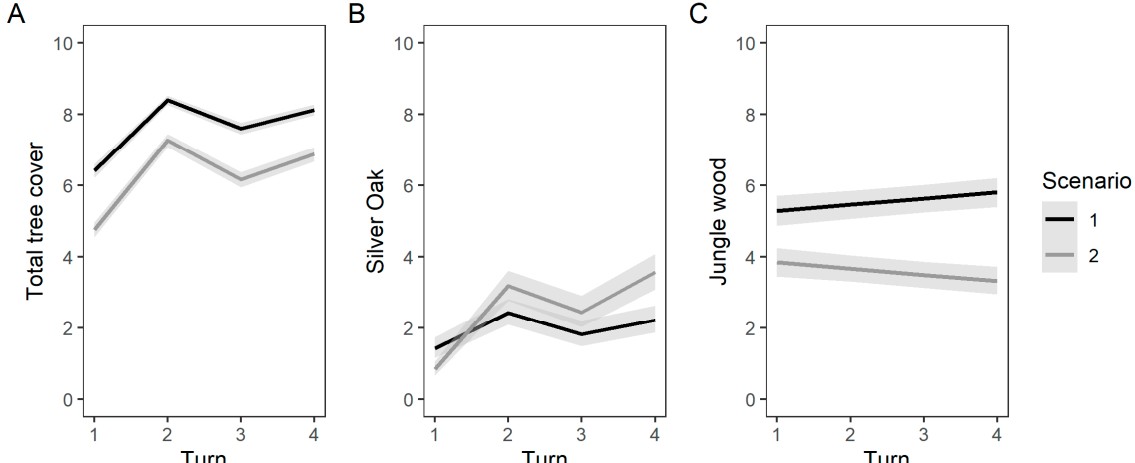

**Figure 5.** Impact of policy scenarios on the tree cover. Evolution per round of (**A**) the total number of trees, (**B**) the number of Silver Oak, and (**C**) the number of Jungle Wood trees at the end of each round on the game board of the players in the two scenarios. In Scenario 1 (black lines), the farmer does not possess the rights to harvest jungle wood and needs to obtain a certificate to cut them, while in scenario 2 (grey lines) no such restrictions are present. For the Silver Oak trade, there are no restrictions in both scenarios. The initial conditions of the game board consisted of six jungle wood trees, two Silver Oaks and two jungle wood seedlings. For all subfigures, the lines represent the predicted mean and the bands indicate the standard errors of these predicted means.

The stand structure shaped by the players in their *Estates* also changed with the scenarios (Figure 6). In each scenario, the structure seems stable after an initial adjustment. Players establish a stable rotation, harvesting only a fraction of their mature *Trees*, thus keeping a constant canopy. However, this "stable state" is very different between scenarios. The proportion of mature *Trees* is significantly lower in Scenario 2 than Scenario 1 (LLR = 231.28, *p* = 0.001).

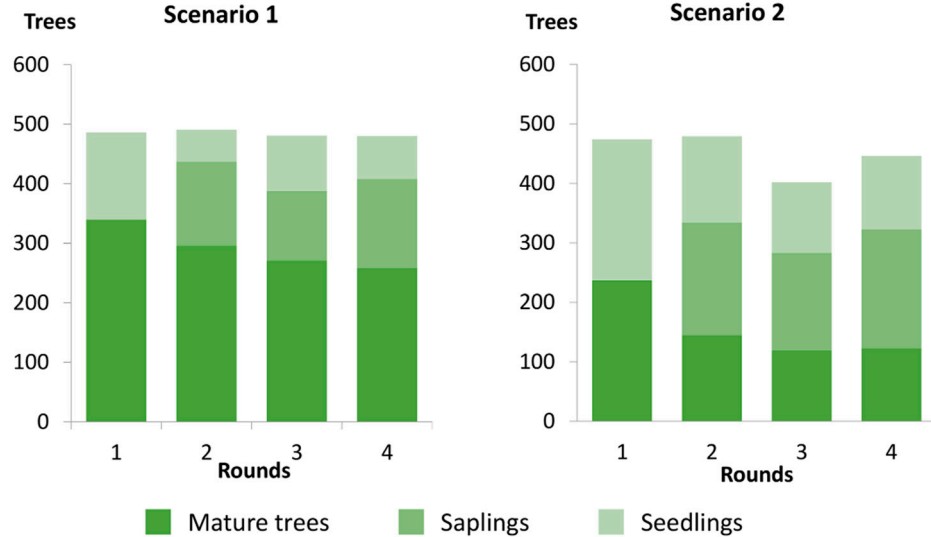

**Figure 6.** Impact of policy scenarios on the canopy structure. The bars represent the age structure of the canopy—seedlings, saplings, mature trees—in proportion of the total number of trees standing. The proportion of mature trees in the gamescape diminishes significantly (S1: 0.54; S2:0.29; z = 7.9; *p* < 0.0001) as they are replaced by seedlings and saplings.

In the second scenario, the proportion exceeded 50% of the *Trees* present in the game, against 30% in the first scenario (see also Figure 5B). When players no longer have restrictions on the cutting and selling of *Jungle Wood*, they increase the pace of transition towards a simplified, *Silver Oak* dominated canopy cover in their *Estates* (Figure 7).

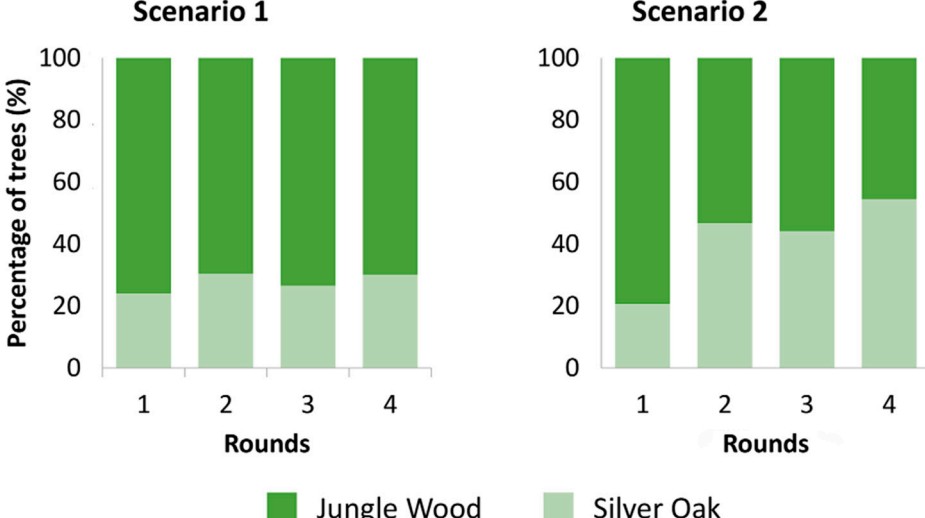

**Figure 7.** Impact of policy scenarios on the canopy composition: Evolution of the proportion of Silver Oak (*Grevillea robusta*) and Jungle Wood trees (saplings and mature) in the gamescape, over the four rounds of both scenarios. The bars represent the proportion of Jungle Wood and *Silver oak*, as a percentage of the total number of trees standing in the gamescape. The proportion of Jungle Wood in the gamescape diminishes significantly in Scenario 2 ($p < 0.0001$), as Players replace it with *Silver oak*.

## 4. Discussion

Our work allows unravelling the different elements weighted by coffee farmers when making decisions about coffee farming, which goes beyond shade management and timber cutting. The purpose of the role-playing game is not to create real world predictions; the behaviour in the game is nothing more than "behaviour in the game" [63]. The results we describe in the "gamescape" do not aim at predicting real policy outcomes, but highlight what could happen should stakeholders behave in a certain way. It thus serves as basis for informed discussions. Our model sacrifices precision for realism and generality [64]: the exact values of the model parameters are meaningless. What matters is that the participants recognize in the causal structure between the elements of the game what they understand to be the causal structure of the elements of the system the game represents. For the process to be useful, the dynamics resulting from the game sessions must "feel real" to the players—all coffee farmers in real life. The strategies they can devise are similar to those of the farmers in the field, and the elements they weight when taking a decision are also part of the model. The starting conditions mimic the current situation in the field. The game presents players with a system where *Silver Oak* represents 20% of the trees in the landscape [41,65]. Other initial conditions could steer the system in other directions, but this would need to be tested and validated. The game parameters are designed in a way that confronts the players with the problem in the time allotted for a game session—a couple of hours. *Trees* grow ten times faster than *Coffee*. The freedom we gain by sacrificing precision does not undermine the model utility. This constrained calibration facilitates dialogue between stakeholders, scientists and officials [51].

We observed that, when given rights, players decide to hasten, rather than reverse, the conversion of the canopy cover from *Jungle Wood* to *Silver oak.* Players respond to the policy scenario by replacing the original cover with one made of a single fast-growing species. This strategy, contrary to what was expected based on the narratives developed by defendants of the farmers' rights, surprised the research team. With the provision of a possible rebound effect on the very first turn, where farmers

would quickly realise the benefits of the new policy change (cutting *Jungle Wood*) before the window of opportunity closed, the behaviour we had come to expect was one of farmers deciding to retain *Jungle Wood* in the system. This should have translated into a stabilisation and/or decrease of the proportion of *Silver Oak* in the last rounds of scenario 2. Based on our results and on subsequent discussions with the players, we have now come to expect the opposite. Simply put, the fast rotation of *Silver Oak* trumps the market premium for high quality native trees, the stated ecosystem services farmers attribute to the original canopy cover, or even a hypothetical existence value of the native trees, all of these arguments having been raised in interviews or during the game sessions. Changing trends towards increased crop monocultures and higher proportions of introduced Silver Oak have also been confirmed from other studies (e.g., [66]).

In addition, our model did not include tree mortality, and, as a result, the only *Jungle Wood trees* removed from the game were the result of a conscious decision by players. We expect the transition to *Silver Oak* would have been even faster in the game had we included tree mortality, as players would probably have made use of the opportunity to replant *Silver oak*. Finally, if we consider that in reality native trees are a mix of many different species, many of them being soft woods without market value [67], the trend we observe in the game is even more conservative.

Our game revealed system components and processes that had been identified in none of the policy narratives of the concerned parties. The existence of informal arrangements with the timber merchants prevent the policies in place from halting the loss of biodiversity. Should farmers receive the rights, the differential growth rate between species would ensure that farmers hasten rather than reverse the trend. These represented hidden pitfalls that would have plunged the coffee-agroforestry system in a non-desired state had the current policy change been implemented as initially designed. This leads to the question of axiology of games and ethics: What if what players learn is bad for the environment, but good for them? (e.g., improved livelihoods through better access to jungle wood and markets, which leads to a faster change of forests with more silver oak). This relates to two key aspects of our gaming approach: responsibility and ethics. As part of the game development, researchers are assumed to be on par with any partaking stakeholders during the game development and gaming process itself [68]. Depending on the objective of our game approach (understanding the system, adapt to its changing conditions if one is less powerful in real life, or change the real life system if one has more power to actually invoke change), the reach of responsibility changes. In the first case (understanding), researchers—including facilitators—ensure the fair and equitable procedure of a game session and its follow-up discussions during the debriefings. Thus, responsibilities remain within the game session that both researcher and stakeholders agreed to partake in. It is up to the organizers to ensure that ethical principles of competence, integrity, responsibility, respect and dignity are maintained throughout the game process. Social learning, or collective learning—i.e., capitalizing on one another's knowledge—is most powerful if people follow a constructive process of argumentation [69,70].

In this process, trust in the model, ownership as participants feel they have contributed to the common description of the world, and momentum to build understanding from one session (e.g., game round) to the other are especially relevant to allow emergence and innovation. Without model elicitation, participants will not trust the model. Without playing the game, participants will not experience the process nor draw any insights. Without the exploration of all possible strategies, decisions made will not be robust to surprises [30]. Each of the three processes is a necessary step, but not sufficient. Integrating the three makes a powerful combination to help stakeholders address the complexities of landscape management.

Landscape dynamics result from the interactions between physical, biological and social processes. We present here an approach that allows stakeholders to understand the drivers of landscape change and explore potential responses. The approach rests on three interacting components. First, an explicit conceptual model that highlights the system boundaries participants agree upon. It is descriptive, and not normative. This includes the ecological processes of the system—the link between shade

and productivity, the differential tree growth between silver oak, and most other tree species found in the coffee agroforestry landscape of Kodagu. Second, a game acts as a boundary object and lets the participants grasp driving forces and explore system behaviour from within. This component is determined by the coffee farmers' needs and aspirations—improving their livelihood, wellbeing and balancing risks. Finally, the definition of scenarios for players to explore the impacts of management interventions. These scenarios define the norms and institutions regulating the access of the latter component to the former—timber regulations, markets, and informal arrangements. Designing the conceptual model builds consensus on how the system works (e.g., [48,71]. Alone, however, this is not enough; the impacts of emergent phenomena and counter intuitive feedback loops cannot be easily predicted from the conceptual model itself. The game, inviting participants to enter the system and be a part of it, gives them increased understanding, even if there is no new data formally introduced. This is an element that requires further research, as we are unable yet to measure the epiphany learning participants report [72]. Scenario testing lets participants explore the resilience of the system. Sessions with identical starting points but divergent outcomes suggest stakeholders have a scope to steer the system in different directions. If game sessions under a specific scenario always reach the same endpoint, the system is probably in a lock-in, and stakeholders are likely powerless to change course and must rely on adapting their strategies instead. Similar structures are found across the planet and are not unique to the case study [73–76]. They constitute the essence of any social and ecological system. Given this complexity, the multiple feedback loops in the system and the capacities of stakeholders to revise their strategies based on new information, anticipating the impacts of a policy change using only one's cognitive capacities is challenging at best, and it is no surprise that policies often fail to meet their objectives.

Games can be used to address conflicts in conservation [49], and the constructivist approach to their use we present here empowers participants. The question then is how outsiders will respond. The process we present will not modify the power structures in place, unless linked to local leadership and legitimacy [77]. Models can help shape better policies, provided the subjectivity of model construction and the limits of their application are duly recognized. The constructivist approach to participatory modelling and the use of games we describe here pave the way for a better representation of human agency in models of landscape change. However, for these models to inspire better policies, practitioners need to give due attention to the theoretical concepts we have presented here.

**Author Contributions:** C.A.G., J.V., N.K., J.K., A.B. and P.V. conceived and designed the experiments; C.A.G., J.V., N.K., J.K., A.B. and P.V. performed the experiments; C.A.G., M.N., A.D., M.D., P.O.W. and C.L.P. analyzed the data; C.A.G., J.V., N.K., J.K., A.B. and P.V. contributed reagents/materials/analysis tools; C.A.G., J.V., K.N., M.N., J.K., A.D., M.D., P.O.W. N.S., A.B., C.L.P., Y.R., R.B., J.G., C.G.K., P.V. wrote the paper. All authors have read and agreed to the published version of the manuscript.

**Acknowledgments:** The research we present here was part of the "Connecting, Enhancing And Sustaining Environmental Services and Market Values of Coffee Agroforestry in Central America, East Africa and India" (CAFNET) project (Europaid/ENV/2006/114–382/TPS). We thank the French Institute of Pondicherry and the Ponnampet College of Forestry for their support, and all the stakeholders that have been participating in the "games" sessions.

**Conflicts of Interest:** The authors declare no conflict of interest.

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
