# Peer review of "Coffee, Farmers, and Trees—Shifting Rights Accelerates Changing Landscapes"

_forests, doi:10.3390/f11040480_

Round 1
Reviewer 1 Report
Abstract
I do not agree that ‘deforestation loss in agroecosystems is generally the result of rational choices, not of a lack of awareness or knowledge’. This sentence does not hold true in the tropics of rural Africa, where farmers have often practiced perennial monocropping agriculture. It is now that some farmers are changing their minds because of adverse effects of global warming on perennial crops survival rates and production
‘Despite both scientific evidence and traditional knowledge that support…’ I doubt that poor smallholders in the tropics of Africa are aware of ‘scientific evidence’
Introduction
Lines 43 to 53: agroecology can also contribute to biodiversity conservation while providing food to farmers
Lines 60-61: please indicate that this holds true in Indonesia
Overall comment: I found this work too virtual and do not see the direct link with conservation of ecosystems. Indeed, there is no evidence that the behavior of players would contribute to make recommendations for conservation. Landscape dynamics are very complex, and I would like that the authors provide evidence that results of the study would contribute to ecosystems conservation.
Author Response
Thank you for your comments and feedback. Please find a point-by-point response to your queries below in this box. In the Attachment, you'll find the revisions in track change of our manuscript.
Best regards,
The authors
Point 1: I do not agree that ‘deforestation loss in agroecosystems is generally the result of rational choices, not of a lack of awareness or knowledge’. This sentence does not hold true in the tropics of rural Africa, where farmers have often practiced perennial monocropping agriculture. It is now that some farmers are changing their minds because of adverse effects of global warming on perennial crops survival rates and production
‘Despite both scientific evidence and traditional knowledge that support…’ I doubt that poor smallholders in the tropics of Africa are aware of ‘scientific evidence’
Response 1: We agree. It is very likely that farmers are not reading scientific publications. The big industries however, they have access to this knowledge pool. The small-scale farmer at the other end of the spectrum is part of another knowledge pool, which he/she is feeding and maintaining. By their very own choice and decision, they will either keep following their habits or change. We believe that the sentence has its merits and validity.
Point 2: Lines 43 to 53: agroecology can also contribute to biodiversity conservation while providing food to farmers
Response 2: True, we are not arguing the pros and cons of agroecology here which is not the purpose of our paper; we are highlighting that humanity with its growing numbers are ipso facto also impacting ecosystems, inter alia, agroforestry systems.
Point 3: Lines 60-61: please indicate that this holds true in Indonesia
Response 3: the paper we cite is reporting on Indonesia. This paragraph is still on global level, hence the international references.
Point 4: Overall comment: I found this work too virtual and do not see the direct link with conservation of ecosystems. Indeed, there is no evidence that the behavior of players would contribute to make recommendations for conservation. Landscape dynamics are very complex, and I would like that the authors provide evidence that results of the study would contribute to ecosystems conservation.
Response 4: Just as a model, the settings of the game provide a selective view of a real-life situation. The way real-life features are selected, i.e. considered as central or anecdotic to analyse the situation, depends on the objective pursued. Hence, the game is a selective approximation of real-life. It does not, in any way, intend to prophesy what will happen in real life. If the game is well-calibrated, it should quickly reveal the crux of the problem to be solved, no matter how the players and/or the hazards of the play will impact its outcome.
In our specific case, we were able to show through the collective behaviour by the players, that the forest landscape would change despite shifting tree rights. This would be a real game changer in conservation if we would have been able to have the results transferred to the forest service and conservation bodies (but this is a problem unrelated to the gaming approach per se, and is rooted in the highly politically explosive situation revolving around land use management in the region).
In brief, our game approach was not primarily to measure participant’s behaviour or to make any predictions about system changes. Our main aim was to use the game to elicit behaviour that is based on individual decisions; together, collective decisions have the power to shift system boundaries and invoke change in the physical landscape. This is what happened in this game; this is what participants have experienced: they have seen changing their (game-)scape. Thus the epiphany of their decisions matter. It is such lessons that can be transferred to real-world management and conservation context.

Reviewer 2 Report
It is a very interesting paper with valid contributions and I therefore recommend that it be accepted for publication after revisions.
The paper starts strong with a good abstract, introduction and materials and methods, but then gets weaker. The results could be a bit richer, going further into the drivers of decision making by coffee farmers. While this is part of the authors’ objective, the results mainly focus on the impact on livelihood and the environment. The information obtained from observing social interactions, documenting the narratives developed by the players and the direct feedback from the participants as mentioned in the material and methods is missing. A main concern is the discussion, as it is lacking in literature support. I therefore advice the authors to revise and improve these sections. The other sections require only minor revisions.
Formatting errors (Error! Reference source not found) and repetition of the figure explanation in the main text, which occurs several times need to be addressed. The quality of the graphs could also be improved.
Please find additional comments on other minor suggestions and recommendations in the attached pdf file.

Author Response
Thank you for your in-depth comments and feedback. Highly appreciated it. Please find below in box a detailed list of our responses to your queries, and in attachment the revised manuscript file (track changes).
We have changed old Figures 5&7; the other figure changes you have requested will follow. We will provide as soon as the responsible author is available --t he editors have approved this. We apologize for this inconvenience.
Best regards
The authors
- Reviewer 2: It is a very interesting paper with valid contributions and I therefore recommend that it be accepted for publication after revisions. ….The paper starts strong with a good abstract, introduction and materials and methods, but then gets weaker. The results could be a bit richer, going further into the drivers of decision making by coffee farmers. While this is part of the authors’ objective, the results mainly focus on the impact on livelihood and the environment. The information obtained from observing social interactions, documenting the narratives developed by the players and the direct feedback from the participants as mentioned in the material and methods is missing. A main concern is the discussion, as it is lacking in literature support. I therefore advice the authors to revise and improve these sections. The other sections require only minor revisions
Authors: The core of the paper is that what matters is not the behaviour exhibited by players, for whatever reasons, but the impact of these behaviours on the system. It is about showing people that their decisions have impacts. That collective decisions can change landscapes. The decisions are not taken on the basis of available information but on the on the basis of expected outcomes. Thus, showing results on social interactions or narratives and player feedback from the debriefings would not add much to this paper as it currently stands. As said, the main interest of the paper was to show that people by strategizing for achieving their own personal goals can leave prints of collective decisions on the physical landscape. We believe that our approach as presented and the results shown here are highlighting the strength of this participatory modelling approach especially to address complex problems such as presented by the Kodagu coffee-agroforest system. Human agency matters and is oftentimes left out in modelling approaches. Here we have put the agency center-stage. We have changed the Discussion section to better reflect exactly this aspect of our research. - Reviewer 2: Formatting errors (Error! Reference source not found)and repetition of the figure explanation in the main text, which occurs several times need to be addressed. The quality of the graphs could also be improved.
Authors: done. - Reviewer 2: L.31: The conceptual model was transformed into a role-playing game and validated by eight workshops with 57 participants …Suggestion "with a total of 57 participants" to make it clear to the readers that it is total number of participants at all 8 workshops
Authors: done (see manuscript file with track changes) - Reviewer 2: L.36: Slow variables is unclear. Please reformulate.
Authors: we have replaced the “slow variables” with “inter alia” - Reviewer 2: L.38: Suggestion: keep it rather general by using India as keyword rather than Kodagu. This should make the paper easier to find.
Authors: done - Reviewer 2: L.80: This last paragraph belongs rather to section 1.3 Objective and Approach.
Authors: we have shifted entire paragraph into 1.3. - Reviewer 2: L.96: (Error! Reference source not found.) please correct
Authors: We have removed this. All these kinds of error messages stem from originally embedded links. We have corrected this throughout the manuscript. - Reviewer 2: Figure 1: this photo is not necassarily relevant as it does not give any additional information. The information from the caption can be included in the text as the pepper vines cannot be seen in the picture anyway.
Authors: We have deleted the picture and caption. - Reviewer 2: L.131: add reference
Authors: we have added the following references: Neilson 2008, Garcia et al. 2010, Leroy et al. 2011 - Reviewer 2: L.132: This section goes too much into details and describes the methods. Consider revising this section to include only the objectives and the main approaches. The other details may be incorporated into the material and methods section.
Authors: We have deleted entire paragraph (“Through workshops and semi structured interviews with coffee farmers….”) since we detail all that later in the Methodology section. - Reviewer 2: L.176: Please express clearly what activities participants make up the total of 72 stakeholders, the 3 group discussions, modeling workshops or 5 game sessions? Perhaps it is best to break it down into the different activities.
Authors: we have added “overall”, since this is the total number of stakeholders for these activities, which means that some of the farmers have been engaging in more than just one of the activities (eg group discussions and gaming). - Reviewer 2: L.178: refer to Figure 2 (which is the conceptual model in question)
Authors: done—we refer now to the new Figure 1 - Reviewer 2: L.183: "Two scenarios were defined, each using the same model but with a slightly different....." Repetition: consider revising … Suggestion: Both scenarios use the same model......
Authors: done - Reviewer 2: L.222: Repetition of caption text for Figure 2. Please delete from main text and refer to the figure where necessary.
Authors: done - Reviewer 2: Figure 2: The text on the arrows are barely legible. Consider increasing font size and using a lighter background to improve visibility. Check colour descriptions: The color of the scripted actors e.g. Coffee Board seems to be red, but it is referred to as orange in the text
Authors: we have relabelled Fig.2 = new Fig. 1. We will provide a remake of original Figures 2, 3, 4, 8, 9 at later stage once responsible author is available. We apologize for this. - Reviewer 2: L.238: check spelling error: "For example, a Coffee Farmers plants ..." …a coffee farmer plants
Authors: done - Reviewer 2: L.244: (Error! Reference source not found.) please correct
Authors: done, reads now Fig. 1 - Reviewer 2: L.246: Repetition: see comment on Fig 2
Authors: done - Reviewer 2: Figure 3: Check labelling in picture: the car seems to be labelled as house. Perhaps explain the formular S=1, T=2 and P=2 on the board.
Authors: We have changed labelling from Fig.3 to new Fig.2 (but see comment 15) - Reviewer 2: L.258: refer to the corresponding Figure
Authors: done; we have rephrased sentences, which read now: “An implicit reality was applied, i.e., a simplification of actors, resources, and space were used in the concept and corresponding game. Thus, the toy model represents an instanciation of the conceptual model. Taken together, the tokens, the board, the rules, the research team, the players and the physical location where the game session is run, constitute an embodiment of the conceptual model presented earlier (Fig. 1)” - Reviewer 2: L.267: You mention in the discussion that this setup represents the current situation in the field. This is relevant to mention here as this is the reason it was designed like this.
Authors: we have added a sentence at the beginning of the paragraph. It reads now: “An implicit reality was applied, i.e., a simplification of actors, resources, and space were used in the concept and corresponding game. Thus, the toy model represents an instanciation of the conceptual model.” - Reviewer 2: L.287: (Error! Reference source not found.)
Authors: done; reads now Fig. 2 - Reviewer 2: Figure 4: see comment on other figures above
Authors: relabelled as Fig. 3 (but see comment 15) - Reviewer 2: L.323: What is meant with: "as the game progresses"? At the second round of each gaming session or only at workshops conducted later? …..Is this direct agreement with the Timber Merchant still legal or is this referred to earlier by the risks of being caught without a permit? However it says that the Timber Merchant does the paperwork for a share, so there should still be a permit involved.
Authors: “as the game progresses” means “as rounds go by”…we have changed this in the manuscript. Regarding the agreement with the timber merchant; this is still an agreement being issued, but the rules have been bent, i.e., a permit has been issued while it actually should not have been issued… so we have seen corruption taking place within the game. - Reviewer 2: L.329: The market crash being announced before round 4 gives the farmers time to react, which normally is not the case. Why announce a market crash before hand?
Authors: it is not announced beforehand.; the game master informs the players that right now there is a market crash happening, which will have consequences on the coffee price. We have changed sentence, which reads now: “In both cases, at the start of the fourth round, a coffee market crash is announced by the game master, which means that the price for coffee bags will drop from 6 to 2 CINR for this turn only.” - Reviewer 2: L.336: How long does a round last? Is there a time limit or how is the end of a round determined?
Authors: There is no time limit. First round takes 1 hour. Others go faster down to 15 minutes. Some time management was done to ensure that both scenarios with five rounds each and debriefings could take place within the scope of a workshop. - Reviewer 2: L.358: If players do not sell their harvest, it is lost" ….Does this refer to coffee or pepper harvest? Coffee is presumably sold to the coffee board rather than other farmers, right?
Authors: We have modified sentence, which reads now: “If players do not sell their harvest, it is lost since there is no storage capacity.” Coffee is sold at the market (not to the coffee board). - Reviewer 2: L.363: Format as text rather than as table description.
Authors: done - Reviewer 2: L.376: For consistency, number equation as done for other equations above.
Authors: We have added an equation number here. - Reviewer 2: L.385: "The third line describes the stochastic component of the model, where the recorded response for the ith observation is drawn from an distribution, g(.) defined by μi and, when the distribution is normal, a standard deviation for the error, σε Models of total income, revenues from coffee, timber and pepper were fitted with linear mixed effects models assuming Gaussian error distributions."….check punctuation
Authors: We have revised this sentence for clarity. - Reviewer 2: L.402: It is perhaps necessary to mention that the participants were sufficiently informed about the research, how their information was treated and that they were free to withdraw their participation at any point, which you already mentioned earlier.
Authors: Consent was demonstrated by the participants attending the events and staying until completion. This sentence already implicitly states this. We have further added this: “Six research assistants of our research team were also member of that local cultural community, and the entire team was therefore aware of possible cultural differences and what is considered proper and respectful in the local context. The only issue that ever arose in the project was people asking why they had not been invited earlier. Important note here: there was no remuneration offered for the half-day time required to participate in the workshop; people from the local communities heard about the game and wanted to be part of it by free will knowing that they had the opportunity to talk freely without fear of any repression by anybody (e.g., authorities, forest service).” - Reviewer 2: L.409: Some parts of this section might fit better in the methods section (especially number of participants in each session and what was measured and recorded).
Authors: We have shift former section 3.1. up and create new section: 2.1.2. Scenario testing - Reviewer 2: L.410: "RPG" This abbreviation is mentioned here for the first time. Please write the full
Authors: done - Reviewer 2: L.421: (values after +/- are standard deviations) is out of place here. It should either be mentioned when presenting the values or in material and methods.
Authors: done; we have deleted it from the original section and placed the sentence where we show the SDs. - Reviewer 2: L.425: (Error! Reference source not found.)
Authors: done - Reviewer 2: L.427: What was the total income? Refer to Figure 5.
Authors: The total income is shown in Fig. 4 a (relabelled the old Fig. 5). We refer to Fig. 4 in this sentence. - Reviewer 2: L433: See comments on other figures…delete this paragraph as it is a repetition of the caption
Authors: done - Reviewer 2: Figure 5: The quality of the graph could be improved and the different graphs could be referred to using unique identifiers to make it easier for the reader.
Authors: We have added identifiers to the panels and redrawn the figures at higher resolution. - Reviewer 2: L.445: (Error! Reference source not found.)
Authors: done; we refer to new Fig. 4 - Reviewer 2: L450: delete this paragraph as it is a copy of the figure caption
Authors: done - Reviewer 2: Figure 6: Consider using different colours as some people are colour blind and might have challenges seeing the differences between red and green. The Quality of the graph could also be improved as some of the text is illegible. …..Why is in S1 the part for "JW Mature Direct Sold" moved out?.....In this graph selling without permit is not mentioned. Please clarify what direct sold means.
Authors: We have deleted this figure as it is redundant; the text says the very same already. - Reviewer 2: L.468: Refer to Figure 7.
Authors: done (see next comment) - Reviewer 2: L.473: (Error! Reference source not found.)
Authors: done; we refer to new Fig. 5 (we have removed old Fig. 6 as it is not adding anything new; text is plenty) - Reviewer 2: L.474: See comments on other figures…delete this paragraph as it is a repetition of the caption
Authors: done - Reviewer 2: Figure 7: Improve quality and use unique identifiers for the different graphs
Authors: Thank you for this suggestion. We have redrawn the figures to increase their quality and added panel identifiers. - Reviewer 2: L.485: Where does left and right refer to?
Authors: We have deleted these and are referring now to respective panels. - Reviewer 2: L.496: See comments on other figures…delete this paragraph as it is a repetition of the caption
Authors: done - Reviewer 2: Figure 8: Improve the quality of the graph. ….Perhaps change the order of the tree development stages. ….Discussing the stability of mature trees, it would improve visual understanding of the development, if mature trees would be on the lowest section in the bar.
Authors: Relabelled to Fig. 6 (but see comment 15) - Reviewer 2: L.506: (Error! Reference source not found.)
Authors: done;we refer to new Fig. 7 - Reviewer 2: L.510: See comments on other figures…delete this paragraph as it is a repetition of the caption
Authors: (see comment 15) - Reviewer 2: Results: Information obtained from observing social interactions, documenting the narratives developed by the players and the direct feedback from the participants as mentioned in the material and methods is not presented here. This would be important in order to get a better understanding the drivers of decision making by coffee farmers.
Authors: See response to comment 1. - Reviewer 2: L.523: The discussion should go deeper, comparing the findings to existing literature.
Authors: We have reworked the Discussion by re-structuring (eg removing “Results-like” portions), by adding further literature/ comparing to other cases/ studies. - Reviewer 2: L.539: First time use of abbreviation
Authors: We have deleted this little paragraph as it is off topic/ beyond the purpose of this manuscript. - Reviewer 2: L.543: Reference
Authors: We have added a reference - Reviewer 2: L.558: Reference
Authors: We have added a reference - Reviewer 2: L.580: Parts of this chapter is still discussing the results rather than only reflecting on the lessons learnt and what should be improved next time. Consider maybe splitting the chapter or renaming it.
Authors: See answer above - Reviewer 2: L.584: Reference
Authors: we did not add reference here since we have deleted this paragraph as it read too much like Results - Reviewer 2: L.599: By sessions do you mean game rounds or workshops? Explain what is meant by session or use a consistent word.
Authors: we have added clarification - Reviewer 2: L.615: Perhaps formulate as recommendation on what could be done differently, rather than saying you failed.
Authors: We have added substantial discussion on axionomy of game instead of dwelling in our failures; thanks for pointing this out - Reviewer 2: L.626: check relevance of some of the information here. …Consider presenting recommendations on possible considerations and what could be done rather than dwelling on what went wrong.
Authors: see comments above

Round 2
Reviewer 1 Report
The authors have addressed the issues that I have raised in the first round; the ms is acceptable for publication
Author Response
Dear Reviewer
Thank you for your comments and recommendation. Please find enclosed updated manuscript.
Best regards
Patrick Waeber
